# Growth Temperature, Trehalose, and Susceptibility to Heat in *Mycobacterium avium*

**DOI:** 10.3390/pathogens9080657

**Published:** 2020-08-15

**Authors:** Simonne Guenette, Myra D. Williams, Joseph O. Falkinham

**Affiliations:** Department of Biological Sciences, Virginia Tech, Blacksburg, VA 24061, USA; sjg6ax@virginia.edu (S.G.); mywillia@vt.edu (M.D.W.)

**Keywords:** *Mycobacterium avium*, growth temperature, trehalose, high temperature survival

## Abstract

*Mycobacterium avium* is capable of an adaptive, reversible response to high-temperature survival depending on its growth temperature. Trehalose concentrations of *M. avium* cells grown at 42 °C were significantly higher compared to those of cells grown at 25 °C. Further, the survival of cells of *M. avium* grown at 42 °C and exposed to 65 °C were significantly higher than the survival of cells grown at 25 °C. This adaptive response to growth temperature may play a role in the persistence of *M. avium* in premise plumbing.

## 1. Introduction

*Mycobacterium avium* is an opportunistic premise plumbing pathogen whose infections in humans have been linked by DNA fingerprinting to the presence of *M. avium* in household plumbing [1,2]. Patients with mycobacterial pulmonary infection are, for the most part, innately susceptible to mycobacteria infection and susceptible to reinfection, even following successful antibiotic therapy [3]. Thus, it would be of value to reduce the exposure to *M. avium* and other nontuberculous mycobacteria (NTM) by reducing their numbers in household plumbing.

In a study of premise plumbing in households across the United States, it was discovered that households whose water heater temperatures were 125 °F (50 °C) or below had nontuberculous mycobacteria (NTM), whereas household plumbing from houses with a water heater temperature of 130 °F (55 °C) or higher seldom had mycobacteria [1]. Based on that background information, a proportion of *M. avium*-infected patients in Wynnewood, Pennsylvania [2] voluntarily raised their water heater temperatures. *M. avium* subsequently disappeared from plumbing samples in 9 of 10 houses by 16 weeks (Lande et al., in preparation). Thus, one way to reduce the numbers of *M. avium* and, presumably, other NTM from household plumbing and prevent reinfection would be to raise water heater temperatures to 130 °F (55 °C).

Despite the observations of *M. avium* behavior in patient households in Wynnewood, Pennsylvania, a concern was that raising water heater temperatures would create conditions for a selection of heat-resistant mutants of *M. avium.* Accordingly, the temperature-resistance of *M. avium* isolates from individual patients and their clonally related household plumbing isolates were measured. Those studies led to the discovery that the sensitivity of *M. avium* to high temperature was influenced by the growth temperature—namely, *M. avium* cells grown at 42 °C, its highest growth temperature, survived exposure to 60 °C for 3 h. That effect was an adaptation, not mutation, as those same household isolates grown at 25 °C were susceptible to 60 °C. As *M. avium* survival to 60 °C was substantially higher than that reported for *M. avium* [4], a study was initiated to describe the growth temperature-mediated adaptation to survival at high temperatures.

A variety of hypotheses offer testable approaches to identify the basis for the differences in heat-susceptibility. First, the *M. avium* strain employed in the study of Schulze-Röbbecke and Bachmann [4] could have been unique and heat-sensitive. Second, in Wynnewood, there has been a selection for heat-resistant mutants among the *M. avium* population in Wynnewood, Pennsylvania over the 20 years since the study of Schulze-Röbbecke and Bachmann [4]. However, those two hypotheses are incompatible with the observation that the high temperature resistance was a reversible adaptation.

One mechanism for adaptive heat resistance in bacteria is through increasing trehalose levels in cells [5,6]. Trehalose is prevalent in members of the genus *Mycobacterium*, and there are three pathways for trehalose synthesis in *Mycobacterium smegmatis* [7]. Mycobacteria are surrounded by both a cytoplasmic and outer membrane, with a true periplasmic space [7]. Trehalose is abundant in the outer membrane and, as glycolipids, are loosely associated with the cell wall and may serve as carriers of mycolic acids from the cells to the outer membrane [7]. Further, trehalose accumulation was linked to the onset of dormancy in *Mycobacterium smegmatis*, and the levels of trehalose correlated with viability measured as growth [8]. 

In this work, we tested the hypothesis that trehalose concentrations in *M. avium* increase as the growth temperature is raised. Further, compared to *M. avium* cells grown at 25 °C, cells grown at 42 °C are more resistant to exposure to heat (i.e., 65 °C).

## 2. Results

### 2.1. Reversibility of Survival Based on Growth Temperature

Cells of *M. avium* strain A5 grown at 42 °C survived three h of exposure to 60 °C (i.e., 75% after 5 min at 60 °C), whereas cells grown at 25 °C did not (i.e., < 1% after 5 min at 60 °C). Reversibility was shown if cells grown at 42 °C were used as an inoculum for a culture grown at 25 °C, as the resulting cells were not resistant to exposure at 60 °C and vice versa. The growth of *M. avium* cells was required for the transition, as shifting a culture grown at 25 °C to a late-log phase at 42 °C for 6–24 h did not increase the survival at 60 °C. Likewise, shifting a late-log phase culture grown at 42 °C to 25 °C for 6–24 h did not reduce the survival at 60 °C.

### 2.2. Trehalose Concentration Increased in M. avium at Higher Growth Temperatures

Following the growth of cells of *M. avium* at different temperatures (i.e., 25 °C, 30 °C, 35 °C, or 42 °C), the cells were collected, broken, and the trehalose and protein concentrations measured in the crude extracts. The protein concentrations were measured to normalize the trehalose values between the samples, as the protein concentrations increased as the growth temperature of *M. avium* increased. The results show that trehalose concentrations relative to protein in *M. avium* strain A5 increased as the growth temperature increased (Table 1).

Only the trehalose/protein ratios of cells grown at 42 °C were significantly higher (*p* < 0.05, Student’s *t*-test of matched pairs) than those of cells grown at 25 °C, 30 °C, or 37 °C. To confirm that this was simply not a unique property of *M. avium* strain A5, the same measurements were performed on crude extracts prepared from cultures of *M. avium* strains Va14 (T) and Va14 (O) isogenic transparent (T) and opaque (O) colony variants (Table 2).

The data confirmed that the increased proportion of trehalose in the cells of *M. avium* strain A5 grown at 42 °C was not a unique feature of that *M. avium* strain. Although the relative concentration of trehalose-to-protein in strain Va14 was lower than that found in strain A5, the same results were obtained with the opaque (O) and transparent (T) colony variants of strain Va14, demonstrating that the effect of the growth temperature was not influenced by colony type. In fact, the increases in the relative concentration of trehalose in strains Va14 (O) and Va14 (T) were higher (i.e., 11.5- and 6.3-fold, respectively) than that measured in strain A5 (7.3-fold).

### 2.3. Trehalose-Protein Ratios in Cellular Fractions

*M. avium* strain A5 cells grown at 25 °C and 42 °C were broken by bead-beating and the crude extract fractionated by ultracentrifugation into a membrane and soluble fraction. The concentrations of trehalose and protein were measured, and the trehalose-to-protein ratios calculated (Table 3). The data showed that significantly (*p* < 0.05, Student’s *t*-test of matched pairs) higher ratios of trehalose-to-protein were found in all fractions from cells grown at 42 °C compared to cells grown at 25 °C, with the highest ratio in the membrane fraction (Table 3).

### 2.4. M. avium A5 Cells Grown at 42 °C are Significantly More Tolerant to 65 °C Than Cells Grown at 25 °C 

Following the growth of *M. avium* strain A5 at 25 °C and 42 °C, the cells were washed and suspended in sterile Blacksburg tap water and exposed to 65 °C for 1−5 minutes, and colony-forming units (CFU) of the surviving cells were measured. The data, expressed as the percent of CFU survival, are presented in Table 4. It can be seen that the percent survivals at 2- and 5-min exposures to 65 °C of *M. avium* strain A5 CFU were significantly (*p* < 0.001) higher for cells grown at 42 °C compared to cells grown at 25 °C (Table 4).

## 3. Discussion

The data included here indicate that a higher growth temperature for *M. avium* results in increased cellular concentrations of trehalose and an increased resistance to high temperatures (i.e., 65 °C). Although increased trehalose concentrations are associated with a high temperature resistance in other microorganisms [5,6], it remains to be determined whether high growth temperature-triggered trehalose synthesis is the determinant of the changes in behavior or is a surrogate for another change, such as the membrane composition. The isolation and characterization of *M. avium* mutants defective in trehalose synthesis would answer that question. It is also possible that the elevated thermal resistance is due to the induction of heat-shock proteins or to a shift in the membrane lipid composition.

The adaptation to a high temperature resistance of *M. avium* is not surprising in light of the fact that the genus *Mycobacterium* includes the waterborne pathogen *Mycobacterium xenopi*, which can grow at 45 °C and survives exposure to 70 °C (i.e., 50% survival after 3 min) [4]. Further, another member of the genus, *Mycobacterium hassiacum*, is a thermophile that grows at 65 °C [9]. 

An adaptation of *M. avium* to a high temperature resistance in response to the growth temperature is consistent with the estuarine habitat of this environmental pathogen [10,11]. In the acidic, black water swamps and in estuaries of the Southeastern Atlantic Coast of the United States, *M. avium* cells will be exposed to high temperatures (e.g., 45−50 °C) during the summer months. In biofilms on rocks, the temperatures in *M. avium* biofilms will likely be quite higher. In addition, the presence of high numbers of *M. avium* in the waters and biofilms of thermal springs (Falkinham, unpublished) also suggests an adaptation to growth and survival at high temperatures.

Evidence of the adaptation of *M. avium* is consistent with its slow growth rate; the generation time is approximately 24 h at 37 °C in a rich laboratory medium. However, *M. avium*’s slow growth rate is not due to a slow rate of metabolism but, rather, due to a diversion of energy (as ATP) to synthesize the long-chain fatty acids (e.g., C_60_–C_80_) that are components of the outer membrane, which makes up 30% of the weight of cells [12]. Thus, even in the face of the inhibition of essential macromolecular synthetic processes, *M. avium* can adapt to changing conditions by inducing activities to overcome that inhibition before irreversible processes lead to cell death.

In addition to speculating about the ecological ramifications of the adaptable, reversible resistance to high temperatures, the results dictate a change in the suggestion to patients to reduce *M. avium* and NTM numbers in household plumbing by raising water heater temperatures. Rather than maintaining a constant water heater temperature of 130 °F (55 °C), the results presented here suggest alternating the water heater temperature between 125 °F (50°) and 130 °F (55 °C), perhaps on a monthly basis.

Further, the results also suggest a novel method for disinfecting hospital heater-coolers that have been identified as the source of cardiac surgery-associated *Mycobacterium chimaera* infections [13]. *M. chimaera*, like its *M. avium* complex (MAC) relative *M. avium* [14], is resistant to chlorine [15] (Falkinham, in preparation). As heater-coolers heat water to 37 °C, it would be expected that *M. chimaera* cells would have increased high-temperature resistance. However, rather than using corrosive chlorine, perhaps a circulation of low-temperature water (e.g., 25 °C) for a sufficient time for *M. chimaera* to adapt by reducing the trehalose concentrations (e.g., 24 h), followed by exposure to 65 °C water for 5 min (Table 4), might reduce *M. chimaera* numbers sufficiently (e.g., 99.9%) to extend the time between repeat disinfection protocols.

## 4. Materials and Methods

### 4.1. M. avium Strains

*M. avium* subspecies *hominissuis* strain A5 is a plasmid-free patient isolate, and *M. avium* strain VA14 and its transparent (T) and opaque (O) colonial variants were employed in this study. Strain VA14 was obtained from the Department of Health Services, Richmond, Virginia.

### 4.2. Growth Media

*M. avium* strains were grown in 25 mL of Middlebrook 7H9 broth (Becton Dickenson, Sparks, MD, USA) containing 0.5% (vol/vol) glycerol and 10% (vol/vol) oleic acid-albumin (M7H9), with aeration (120 rpm) at different temperatures (25 °C, 30 °C, 35 °C, or 42 °C) to a mid-log phase in 500-mL Nephalometer flasks. Growth, as turbidity (abs 540 nm), was measured and plotted to identify the growth phases. *M. avium* cells were enumerated as colony-forming units (CFU) on Middlebrook 7H10 agar (Becton Dickenson, Sparks, MD, USA) containing 0.5% (vol/vol) glycerol and 10% (vol/vol) oleic acid–albumin (M7H10).

### 4.3. Water Acclimation

Cells grown to the log phase were collected by centrifugation (5000× *g*) for 20 min, the supernatant medium discarded, and the cells suspended in the same volume of sterile Blacksburg tap water. The washed cells were pelleted by centrifugation, the supernatant discarded, and the cells suspended in the same volume of sterile Blacksburg tap water. The twice-washed cells were incubated for 1 week at room temperature, with aeration (60 rpm) in 500-mL flasks. Following water acclimation, CFU of 10-fold serial dilutions in sterile Blacksburg tap water were measured on M7H10 agar. For some experiments, the water-acclimated cells were concentrated 10-fold by centrifugation (5000× *g* for 20 min) and suspension of the pelleted cells in one-tenth the volume (25 mL) of sterile deionized water.

### 4.4. Isolation of Crude, Membrane, and Soluble Fractions

Cells were fractionated as described [16]. Briefly, following the growth of cultures, cells from 25 mL were collected by centrifugation (5000× *g* for 20 min), the supernatant medium discarded for autoclaving, and cells suspended in 25 mL of distilled water. The washed cells were again collected by centrifugation (5000× *g* for 20 min), suspended in 2 mL of distilled water, and broken by “bead-beating” (0.1-mm-dia. beads, Mini-Bead Beater, Omni International, Kennesaw, GA, USA). The broken cell suspension was centrifuged at 5000× *g* for 10 min to pellet unbroken cells and the supernatant “crude extract” collected. A portion of the “crude extract” was centrifuged at 50,000× *g* for 60 min and the supernatant “soluble fraction” collected. The pellet was suspended in 1 mL of distilled water to yield the “membrane fraction”.

### 4.5. Measurement of Protein Concentration in Cell Lysates

Total protein in the broken cell suspensions (crude extract) and soluble and pellet fractions was measured using the method of Lowry et al. [16].

### 4.6. Measurement of Trehalose Concentrations in Cells

Trehalose concentrations in the broken cell suspensions and fractions were measured as described [6]. Briefly, trehalose in broken cell suspensions or fractions was converted enzymatically to glucose by the action of Trehalase (porcine kidney, Sigma-Aldrich, St. Louis, MO, USA) and the glucose concentration measured using the glucose oxidase peroxidase-coupled assay (glucose, GO) Assay Kit (Sigma-Aldrich, St. Louis, MO, USA) against a standard curve. The glucose concentrations of cell lysates and fractions were measured before the addition of the enzyme Trehalase to correct for endogenous levels of glucose before trehalose transformation. 

### 4.7. Measurement of Heat Susceptibility

Cells of *M. avium* strain A5 were grown at 25 °C, 30 °C, 35 °C, or 42 °C, as described above, and cell suspensions were exposed to 65 °C and surviving cells measured as colony-forming units (CFU). Cells from 25 mL of the mid-log phase cultures grown at the different temperatures were collected by centrifugation (5000× *g* for 20 min), the supernatant medium discarded for autoclaving, and the cells suspended in 25 mL of sterile Blacksburg tap water. One milliliter of each suspension was transferred to a sterile microfuge tube and placed in a water bath set at 65 °C. Immediately, the number of colony-forming units (CFU)/mL was measured by withdrawing 0.1 mL and mixing in 9.9 mL of sterile, distilled water (100-fold dilution). A 10-fold dilution series was prepared, and 0.1 mL of each were spread on M7H10 agar in triplicate. After 1, 2, and 3 h of exposure to 65 °C, 0.1-mL samples were withdrawn and a 10-fold dilution series prepared, and 0.1 mL of each dilution was spread on M7H10 agar plates in triplicate. Plates were sealed with Parafilm™ (Amcor, Zurich, Switzerland), incubated at 37 °C for 10–14 days to allow for colony appearance, and colonies counted to calculate the CFU/mL at each time point—namely, 0, 1, 2, and 3 h of exposure. Survival values at 1, 2, and 3 h of exposure to heat were calculated based on the CFU/mL of the immediate (control) sample.

### 4.8. Statistical Analysis

Differences in the measurements for *M. avium* strain A5 cells grown at 25 °C and 42 °C were assessed statistically using the Student *t*-test for matched populations using GraphPad InStat®, Version 3.0 Software (San Diego, CA, USA). 

## Figures and Tables

**Table 1 pathogens-09-00657-t001:** Relative trehalose concentrations in crude extracts of *Mycobacterium avium* strain A5 grown at different temperatures.

Growth Temperature	Trehalose/Protein in Crude Extract ^a^
25 °C	1.43 ± 0.67 (6)
30 °C	1.55 ± 0.37 (2)
35 °C	2.19 ± 0.77 (2)
42 °C	10.47 ± 0.47 (4)

^a^ Average of duplicate measurements of (n) independent cultures.

**Table 2 pathogens-09-00657-t002:** Relative trehalose concentrations in crude extracts of *M. avium* strains Va14 (T) and Va14 (O) grown at 25 °C and 42 °C.

Growth Temperature	Trehalose/Protein in Crude Extract ^a^	
	Va14 (O)	Va14 (T)
25 °C	0.21 ± 0.08	0.21 ± 0.10
42 °C	2.41 ± 0.42	1.32 ± 0.46

^a^ Average of duplicate measurements of two independent cultures.

**Table 3 pathogens-09-00657-t003:** Trehalose distribution in crude extract, membrane, and soluble fractions ^a^.

Growth Temperature	Trehalose/Protein Ratio ^a^		
	Crude Extract	Membrane Fraction	Soluble Fraction
25 °C	1.85	1.63	1.25
42 °C	11.05	12.01	11.34

^a^ Average of duplicate measurements of two independent experiments.

**Table 4 pathogens-09-00657-t004:** Survival at 65 °C of *M. avium* cells grown at 25 °C and 42 °C.

Growth Temperature	Percent Survival at 65 °C ^a^		Trehalose/Protein ^a^
	2 min	5 min	Ratio
25 °C	2.0 ± 0.4%	0.0028 ± 0.0003%	1.8 ± 0.6
42 °C	9.4 ± 0.4%	0.26 ± 0.01%	3.1 ± 0.5

^a^ Average of duplicate measurements of two independently grown cultures.

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
