# Peer review of "Growth Temperature, Trehalose, and Susceptibility to Heat in Mycobacterium avium"

_pathogens, 2020, doi:10.3390/pathogens9080657_

Round 1
Reviewer 1 Report
The introduction is excellent and frames the study along with a strong rationale for conducting the work.
The results are logical and well done. In some cases, graphs might display the data better than tables, but that is subjective.
The primary concern with the work is that other factors, such as heat shock protein expression, could be contributing to the heat resistance. This is especially likely when cells are primed at 42C prior to exposure to the higher temps. The authors hint at possibilities other than trehalose in the first paragraph of the discussion, but heat shock proteins should be explicitly mentioned.
Another concern is the way the trehalose concentration was measured. It appears that glucose was actually measured based on an enzymatic assay that converts trehalose to glucose. But was endogenous glucose measured for background in these same cells?
Minor comments:
The A5 strain is not well defined in this study. Does it have an IS1245 RFLP type? Is the subspecies of M. avium known? It would help interpretation if the strain had more details than simply A5.
Section 2.1 heading should be changed to “Reversibility of SURVIVALbased on growth temperature”.
Author Response
Reviewer 1.
I appreciate the positive comments and thoughtful suggestions.
- The manuscript has been revised in the first paragraph of the Discussion to include the possibility that the adaptive thermal resistance could be due to either induction of heat-shock proteins or a shift in the composition of membrane lipids.
- The Materials and Methods, Section 4.6. Measurement of Trehalose Concentrations in Cells has been revised to list that glucose concentrations of cell lysates and fractions were measured before the addition of the enzyme trehalase to account for the contribution of indogenous glucose.
- Strain Mycobacterium avium A5 has been further described as a subspecies hominissuis patient isolate.
Reviewer 2 Report
I enjoyed reading the manuscript you prepared.
I find the topic very interesting from a practical point of view. Thermoresistance of microorganisms is a growing problem and in my opinion it is important to look for methods to reduce it. The results obtained in the experiment prove that the differentiation of the water heating temperature can effectively eliminate the problem of the formation of thermoresistant bacterial strains (not only Mycobacterium avium) and allows to reduce the amount of disinfectants used in the water treatment process.
The subject of the experiment was M. avium, which, like other slowly growing mycobacteria, is characterized by a specific structure of the cell wall. I think it would be worth introducing the reader to the structure of the mycobacterial cell wall, taking into account the role of Trehalose present in it in building thermal resistance. This would make it easier for a reader unfamiliar with mycobacteria to understand the purpose of the research.
line 32 - is it really 130 degrees Celsius?
Author Response
Reviewer 2.
The authors thank this Reviewer for their gracious comments and understanding of how the observations can help prevent the selelction of thermoresistant bacterial variants in water heaters.
- The Introduction (paragraph 5) has been revised to include more information on the structure of the cell envelope of mycobacteria and the role of trehalose. Specifically, "Mycobacteria are surrounded by both a cytoplasmic and outer membrane with a true periplasmic space [7]. Trehalose in abundant in the outer membrane and ,as glycolipids, are loosely associated with the cell wall and may serve as carriers of mycolic acids from the cells to the outer membrane [7]."
- The mistake in the water heater temperature has been corrected to 130 F (55 C).
Round 2
Reviewer 1 Report
No comments